# Stress-mediated exit to quiescence restricted by increasing persistence in CDK4/6 activation

Hee Won Yang[1,2†]*, Steven D Cappell[1,3†], Ariel Jaimovich[1†], Chad Liu[1], Mingyu Chung[1], Leighton H Daigh[1], Lindsey R Pack[1], Yilin Fan[1], Sergi Regot[4,5], Markus Covert[1,5], Tobias Meyer[1,6]*

[1]Department of Chemical and Systems Biology, Stanford University School of Medicine, Stanford, United States; [2]Department of Pathology and Cell Biology, Columbia University Medical Center, New York, United States; [3]Laboratory of Cancer Biology and Genetics, Center for Cancer Research, National Cancer Institute, Bethesda, United States; [4]Department of Molecular Biology and Genetics, Johns Hopkins University School of Medicine, Baltimore, United States; [5]Department of Bioengineering, Stanford University School of Medicine, Stanford, United States; [6]Department of Cell and Developmental Biology, Weill Cornell Medical College, New York, United States

*For correspondence:
hy2602@cumc.columbia.edu (HWY);
tom4003@med.cornell.edu (TM)

†These authors contributed equally to this work

Competing interests: The authors declare that no competing interests exist.

**Abstract** Mammalian cells typically start the cell-cycle entry program by activating cyclin-dependent protein kinase 4/6 (CDK4/6). CDK4/6 activity is clinically relevant as mutations, deletions, and amplifications that increase CDK4/6 activity contribute to the progression of many cancers. However, when CDK4/6 is activated relative to CDK2 remained incompletely understood. Here, we developed a reporter system to simultaneously monitor CDK4/6 and CDK2 activities in single cells and found that CDK4/6 activity increases rapidly before CDK2 activity gradually increases, and that CDK4/6 activity can be active after mitosis or inactive for variable time periods. Markedly, stress signals in G1 can rapidly inactivate CDK4/6 to return cells to quiescence but with reduced probability as cells approach S phase. Together, our study reveals a regulation of G1 length by temporary inactivation of CDK4/6 activity after mitosis, and a progressively increasing persistence in CDK4/6 activity that restricts cells from returning to quiescence as cells approach S phase.

## Introduction

Mitogens promote entry into the cell cycle in part by inducing the expression of cyclin Ds to activate CDK4 and its paralog CDK6 (CDK4/6) (*Matsushime et al., 1994*). A main role of CDK4/6 activation is to phosphorylate retinoblastoma protein (Rb), which is inactivated by hyperphosphorylation on approximately 15 sites (*Dick and Rubin, 2013*; *Topacio et al., 2019*). Unphosphorylated or mono-phosphorylated Rb proteins inhibit chromatin-bound E2F (mostly E2F1-3), repressing the E2F-mediated expression of a large set of cell-cycle regulators including cyclin Es and cyclin As (*Dick and Rubin, 2013*; *Narasimha et al., 2014*; *Nevins, 2001*). When hyperphosphorylated, Rb dissociates from chromatin-bound E2F, promoting entry into the cell cycle by a progressive increase in the activity of CDK2 (*DeGregori et al., 1995*; *Spencer et al., 2013*), and inactivation of the anaphase-promoting complex/cyclosome-Cdh1 (APC/C^Cdh1) shortly before cells enter S phase (*Cappell et al., 2016*; *Grant et al., 2018*; *Ondracka et al., 2016*).

While it is well established that E2F-mediated expression of cyclin E and A promotes activation of CDK2 to drive entry into S-phase, there are conflicting findings about the role of CDK4/6, including: (i) how CDK4/6 and CDK2 cooperate to regulate hyperphosphorylation of Rb and thus E2F gene expression, and (ii) how CDK4/6 is activated. Early studies proposed that CDK4/6 activity may only partially phosphorylates Rb while a CDK2-activity driven positive feedback loop subsequently hyperphosphorylates Rb (*Geng et al., 1996*; *Zetterberg et al., 1995*). Two other studies concluded that CDK4/6 activity only monophosphorylates Rb and E2F targets remain suppressed unless Rb is hyperphosphorylated by CDK2 (*Narasimha et al., 2014*; *Sanidas et al., 2019*). Our group reported that CDK4/6 activity can be sufficient to hyperphosphorylate Rb in G1, since mitogens still trigger hyperphosphorylation of Rb in mouse embryonic fibroblasts (MEFs) where all four cyclin E and A genes were deleted. Furthermore, there are conflicting results whether sufficient active cyclin D-CDK4 dimers are present in cells to phosphorylate Rb, and whether the relevant cyclin D-CDK4/6 activity requires binding of the CIP/KIP CDK inhibitors p21 or p27. Such trimeric CDK4/6 complexes can be active (*Sherr and Roberts, 1999*), and tyrosine phosphorylation of p27 can generate active trimeric CDK4/6 complexes (*Blain, 2008*; *Guiley et al., 2019*), but studies using double p21/p27 (*Cheng et al., 1999*) and triple p21/27/p57 (*Tateishi et al., 2012*) knockout cells came to different conclusions whether binding of CIP/KIP type CDK inhibitors is required for cells to contain active cyclin D-CDK4/6. Addition of the cyclin D-CDK4/6 selective inhibitor palbociclib in late G1 also caused dephosphorylation of hyperphosphorylated Rb in less than 15 min (*Chung et al., 2019*), while an active cyclin D-CDK4 complex with bound tyrosine phosphorylated p27 was unresponsive to palbociclib inhibition (*Guiley et al., 2019*), raising additional questions how CDK4/6 activity is regulated in cells.

Such open questions regarding CDK4/6 activity motivated us to develop a CDK4/6 activity reporter. We particularly considered that a combined CDK4/6 and CDK2 activity reporter system could be used along with genetic, mitogen, stress, and pharmaceutical perturbation experiments to provide an alternative approach to reconcile conflicting results and answer open questions. We previously developed a nuclear translocation-based reporter that can monitor the activation of cyclin E-CDK2 in G1 phase (*Hahn et al., 2009*; *Spencer et al., 2013*) and different properties of the reporter were characterized in subsequent studies. The reporter can be phosphorylated in vitro by cyclin E-CDK2 or cyclin A-CDK2 activity (*Spencer et al., 2013*), as well as by cyclin E/A-CDK1 activity (*Schwarz et al., 2018*), but not by cyclin D-CDK4/6 activity (*Spencer et al., 2013*). Given that cyclin E prefers CDK2 over CDK1 (*Koff et al., 1992*), and that cyclin A typically starts to increase at the G1/S transition, this cyclin E/A-CDK2/1 reporter is expected to primarily measure the activity of cyclin E-CDK2 during G1 phase. We therefore refer to the reporter here as a 'CDK2 reporter'.

An unexpected result of CDK2 reporter measurements was that there is great variability in the time course of CDK2 activation during G1 between individual cells in the same population (*Barr et al., 2017*; *Schwarz et al., 2018*; *Spencer et al., 2013*). Different cells activate CDK2 activity at different times, or not at all, and there is also great variation when cells enter S phase. Thus, the population heterogeneity in cultured cells makes it challenging to make conclusive interpretations from biochemical data about the single-cell time course of cyclin D-CDK4/6 and CDK2 activation and Rb hyperphosphorylation. The variability in the CDK2 activity time-courses was a second motivation to develop a reporter for CDK4/6 activity to determine when individual cells activate CDK4/6 relative to CDK2 activation and S phase entry.

Mitogen stimulation leads to a gradual increase in cyclin D expression over a period of many hours (*Baldin et al., 1993*) and one would therefore expect that the increase in cyclin D is paralleled by a gradual increase in CDK4/6 activity. Nevertheless, whether CDK4/6 is activated with a gradual or rapid kinetics and if and when CDK4/6 is activated when cells exit mitosis remained open questions. It is often believed that CDK4/6 activation is a main trigger activity for cell-cycle entry in cells exiting quiescence as well as in cycling cells (*Bertoli et al., 2013*; *Meyerson and Harlow, 1994*), and that CDK4/6 activation is only later followed by cyclin E-CDK2 activation, APC/C$^{Cdh1}$ inactivation, and S-phase entry. However, given the heterogeneity when individual cells enter S phase and the gradual increase in cyclin D expression, such an orderly progression with CDK4/6 being always activated first could not be definitely answered without a CDK4/6 reporter. Most importantly for our study here, recent studies showed that stress signals can reverse cell-cycle entry even after cyclin E-CDK2 has already been activated until APC/C$^{Cdh1}$ is inactivated at the onset of S phase (*Cappell et al., 2016*; *Heldt et al., 2018*). However, it was not known whether stress signals act by

first inactivating CDK4/6 and what the relationship is between a potential stress-mediated inactivation of CDK4/6 and CDK2 in G1 phase.

Here, we developed a reporter system to measure CDK4/6 and CDK2 activities in live single-cells to gain initial insights and help future studies to reconcile some of these conflicting results and answer open questions. In the MCF10A cell model that we used to evaluate the usefulness of the reporter, we show that CDK4/6 activity stays low for a long and variable time period after mitogen stimulation of quiescent cells, but then rapidly increases, consistent with an ultrasensitive CDK4/6 activation mechanism. Activation of CDK4/6 is then reliably followed by CDK2 activation and a delayed inactivation of APC/C$^{Cdh1}$. In cycling cells exiting mitosis, CDK4/6 can already be active and Rb hyperphosphorylated in a subset of cells, and these cells start to immediately increase their CDK2 activity and continue on to enter the next cell-cycle after a short G1. In contrast, cells in the same population can have low CDK4/6 and CDK2 activity as well as low Rb phosphorylation after mitosis. Some of these cells keep CDK4/6 activity low and exit to quiescence, while the remaining cells again rapidly increase CDK4/6 activity after a variable delay to enter the next cell-cycle, explaining how cells can generate a variable G1 length. When we applied different types of stresses to cells with active CDK4/6 and CDK2 in G1, we found that stress signals can rapidly inhibit CDK4/6 while inhibiting CDK2 activity only after a delay, demonstrating that CDK4/6 activity is both a primary target for mitogen as well as stress signaling. Finally, our study supports a principle of progressive persistence during G1 phase whereby the probability for cells to respond to stress and inactivate CDK4/6 becomes progressively smaller as cells get closer to S phase.

## Results

### Design and validation of a fluorescent reporter system to monitor CDK4/6 and CDK2 activities

We used a kinase translocation reporter (KTR) strategy developed for MAPK family kinases (*Regot et al., 2014*) to construct a reporter that is sensitive to changes in CDK4/6 activity (*Figure 1A*). In this reporter design, the relative nuclear localization of the reporter is controlled by competing nuclear import and export sequences regulated by kinase-specific phosphorylation. Specifically, the putative CDK4/6 reporter is comprised of a fluorescent protein (mCherry) conjugated to a peptide that contains nuclear import and nuclear export sequences with residues suitable for phosphorylation by CDKs. Relative selectivity for CDK4/6 activity was achieved by adding a cyclin D-CDK4/6-specific docking site fused at the C-terminus. The CDK4/6-specific docking site is a short C-terminal fragment of Rb (aa 886–928) that has previously been shown to be required for binding of CDK4 to Rb and also required for the phosphorylation of Rb by cyclin D-CDK4 but not by cyclin E-CDK2 (*Topacio et al., 2019*; *Wallace and Ball, 2004*; *Figure 1—figure supplement 1A*). By co-expressing the reporter along with a fluorescent nuclear histone marker, one can derive a putative CDK4/6 activity by measuring the relative localization of the reporter between the cytoplasm and nucleus at the single-cell level using automated imaging as previously described for the CDK2 reporter (*Spencer et al., 2013*; *Figure 1A*). To validate the reporter and test possible uses, we established human epithelial MCF10A cells expressing a nuclear marker, the putative CDK4/6 reporter, a reporter for CDK2 activity (*Hahn et al., 2009*; *Spencer et al., 2013*), as well as a reporter to measure the inactivation of APC/C$^{Cdh1}$ before the G1/S transition (*Cappell et al., 2016*; *Sakaue-Sawano et al., 2008*; *Figure 1A,B*).

We first monitored the putative CDK4/6 reporter in single cells to test for a change in nuclear localization after mitogen stimulation of serum-starved cells (*Figure 1C*). The reporter translocated from the nucleus to the cytoplasm before CDK2 activity starts to increase and APC/C$^{Cdh1}$ is inactivated (*Figure 1D* and *Figure 1—figure supplement 1B*). We next tested whether the reporter shows relative specificity for CDK4/6 over CDK2/1 activity. To test for specificity, we treated cells with different CDK inhibitors 11 hr after mitogen stimulation and selected cells where the signal from the putative CDK4/6 and CDK2 reporters have both increased before drug treatment. Additionally, we classified cells as being in G1 or S phase based on the absence or presence of a signal from the APC/C$^{Cdh1}$ reporter construct. Application of a CDK1 selective inhibitor (RO-3306) did not alter the putative CDK4/6 reporter signal in G1 or S (*Figure 1—figure supplement 2*), suggesting that the reporter does not measure an increase in CDK1 activity for the conditions tested. The CDK1

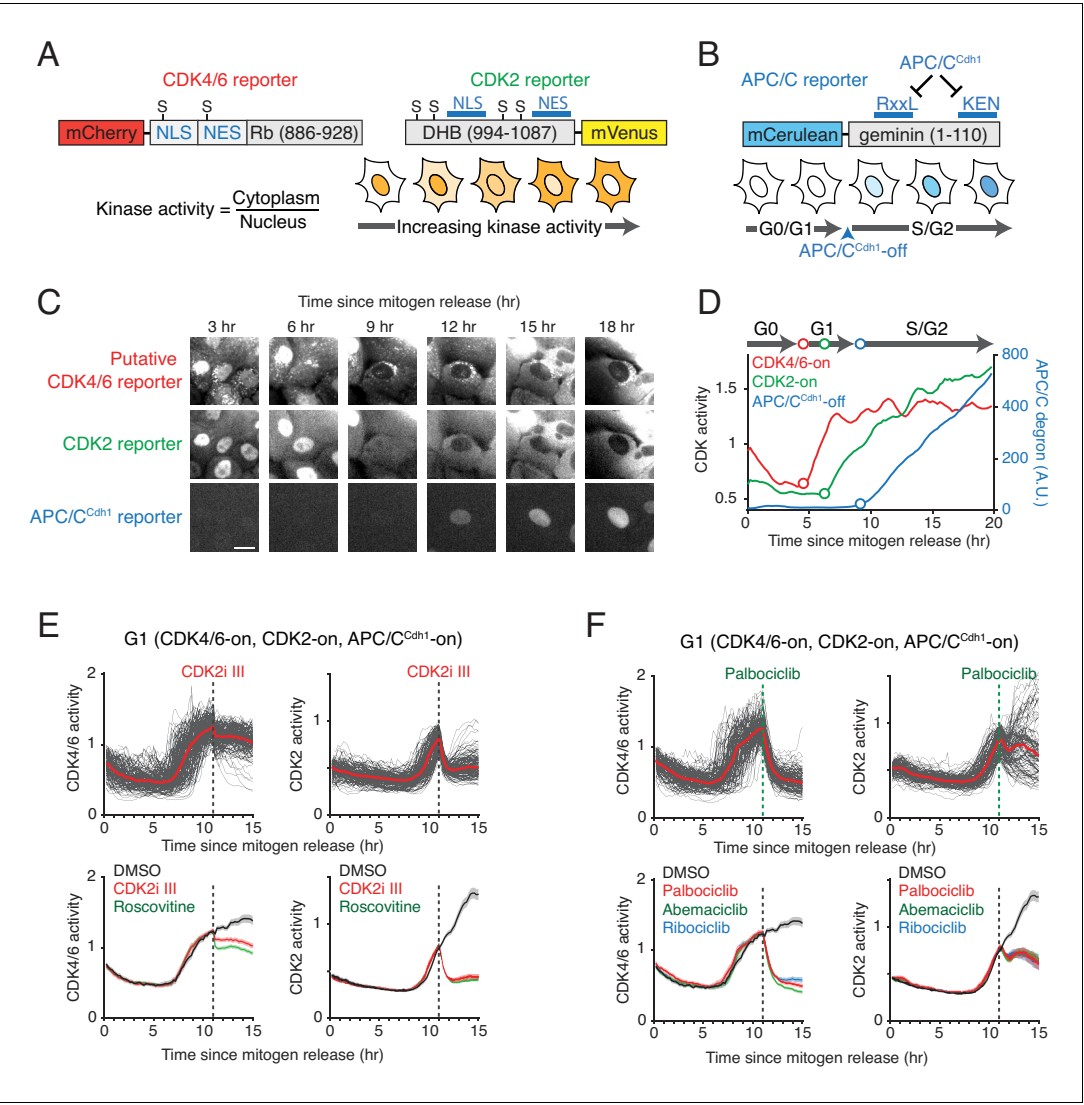

**Figure 1.** Characterization of the selectivity of the putative CDK4/6 reporter. (A) Schematics of CDK4/6 and CDK2 reporters. Increased CDK kinase activity increasingly phosphorylates a larger fraction of the respective reporter and promotes an increased relative cytoplasmic versus nuclear localization of the reporter. (B) Schematic of the APC/C$^{Cdh1}$ reporter. Once APC/C$^{Cdh1}$ is inactivated at the G1/S transition, the intensity of the reporter starts to gradually increase. (C) Example of a time series of images of the three reporters used for automated analysis of the respective signaling activities in the same cell. Scale bar is 10 μm. (D) An example of three cell-cycle activity time courses corresponding to the cell in the center of the images in (C). (E, F) Response of the two CDK reporters in G1 phase to (E) CDK2 inhibitors, CDK2i III (60 μM) and Roscovitine (60 μM), and (F) CDK4/6 inhibitors, Palbociclib (1 μM), Abemaciclib (1 μM), and Ribociclib (1 μM). Multiple single-cell time courses are shown (top; red line indicates median value). Median values and confidence intervals are also shown (bottom: shade indicates 95% confident interval) (n > 90 cells). Only cells with activated CDK4/6 and CDK2 were selected. Inhibitors were added at 11 hr after mitogen release (dotted line).

The online version of this article includes the following figure supplement(s) for figure 1:

**Figure supplement 1.** Design of CDK4/6 reporter and examples of single-cell traces of CDK4/6, CDK2, and APC/C reporter signals after mitogen release.
**Figure supplement 2.** Response of CDK4/6 and CDK2 reporters to CDK inhibitors.
**Figure supplement 3.** Knockout control experiments of cyclin Es and As to exclude the possibility that CDK2 plus CDK4/6 activity is required for generating a CDK4/6 reporter signal.
**Figure supplement 4.** Derivation of a corrected CDK4/6 activity using simultaneously measured CDK2 activity to subtract a contribution to the reporter signal from CDK2 activity.

inhibitor also did not have an effect on the CDK2 reporter in G1 phase, supporting the previous hypothesis that the measured CDK2 signal in G1 is primarily from cyclin E-CDK2, as cyclin A is expected to be degraded in G1 by APC/C$^{Cdh1}$ (*Figure 1—figure supplement 2A*). We next tested for the relative selectivity of the two reporters using roscovitine, which has similar inhibitory selectivity for cyclin B-CDK1 and cyclin E/A-CDK2 kinase complexes (*McClue et al., 2002*), and CDK2i III (CVT-313), a kinase inhibitor with a reported 8-fold selectivity for inhibiting cyclin E/A-CDK2 over cyclin B-CDK1 (*Brooks et al., 1997*). Both drugs show in vitro minimal inhibition of cyclin D-CDK4/6 activity. In addition, we used the clinically approved CDK4/6 inhibitors (Palbociclib, and Ribociclib) that have only a minimal effect on cyclin-CDK2/1 complexes, and another approved CDK4/6 inhibitor (Abemaciclib) that is less selective for cyclin D-CDK4/6 and can also inhibit several additional kinases (*Hafner et al., 2019*). As expected, addition of CDK2i III and roscovitine significantly suppressed cyclin E-CDK2 activity during G1 phase while also reducing the signal from the putative CDK4/6 reporter by about 10–20% during that time (*Figure 1E*). Conversely, each of the three CDK4/6 selective inhibitors nearly fully inhibited the putative CDK4/6 reporter in G1 phase (*Figure 1F*). We note that subsets of G1-phase cells slowly reversed the normal steady increase in cyclin E-CDK2 activity following CDK4/6 inhibition, consistent with CDK4/6 still contributing to the activation of cyclin E-CDK2 in G1. Together, these control experiments suggest that the putative CDK4/6 reporter is selective for measuring CDK4/6 over CDK2/1 activity in G1.

We next tested for the suitability of the reporter to measure CDK4/6 activity in S or G2 phase when CDK2 activity is higher compared to G1 phase. When CDK4/6 activity is acutely inhibited in S/G2, approximately 35% of the putative CDK4/6 reporter signal cannot be suppressed by the CDK4/6 inhibitors (*Figure 1—figure supplement 2B*). As an additional control, we measured the effect of a few additional protein kinase inhibitors (*Figure 1—figure supplement 2C*). Together, these results suggest that, particularly at higher levels of CDK2 activity during S and G2 phase, CDK2 activity is likely a main additional contributor to the measured reporter signal but most of the measured signal can still be inhibited by addition of CDK4/6-selective inhibitors.

Given that there is a contribution from CDK2 activity to the CDK4/6 reporter signal, we determined whether there might be a co-requirement for CDK2 plus CDK4/6 activity, meaning that CDK4/6 activity alone may not be sufficient on its own to generate a CDK4/6 reporter signal. Such a co-requirement for the two kinases has for example been proposed for Rb hyperphosphorylation (*Zetterberg et al., 1995*). Notably, a recent study by our group showed that CDK4/6 activity can still hyperphosphorylate Rb when all cyclin As and Es genes are missing in MEFs, suggesting that CDK4/6 activity alone can be sufficient for hyperphosphorylating the substrate Rb (*Chung et al., 2019*). We tested for a potential co-requirement of cyclin E/A-CDK2 activity in regulating the putative CDK4/6 reporter activity by also utilizing a Cre-Lox cell line to conditionally knockout all four cyclin E and A genes (*Figure 1—figure supplement 3A*). After induction of Cre, the CDK2 reporter signal in MEFs without cyclin E/A was, as expected, indistinguishable from basal, confirming that the CDK2 reporter signal requires cyclin E or A (*Figure 1—figure supplement 3B*). Importantly, the putative CDK4/6 reporter signal was still observed in the cyclin E/A knockout MEFs, and acute CDK4/6 inhibition still suppressed the putative CDK4/6 reporter signal (*Figure 1—figure supplement 3C*). Together, these control and knockout measurements argue that CDK4/6 activity alone can generate a CDK4/6 reporter signal, without that E or A type cyclins or CDK2/1 activity are needed for the CDK4/6 reporter signal to increase.

Nevertheless, one is still left with an additive contribution from CDK2 activity that is responsible for part of the measured CDK4/6 reporter signal particularly in S and G2 phase. Since the dual reporter system measures both the CDK4/6 and the CDK2 reporter signal in the same cell, it is possible to derive a corrected CDK4/6 activity by adjusting for the additive contribution from CDK2 activity. This can be achieved by subtracting a 35% fraction of the CDK2 reporter signal from the simultaneously measured CDK4/6 reporter signal in the same cell to derive a corrected CDK4/6 activity throughout the cell cycle (see Methods). When this correction is applied, the corrected CDK4/6 activity signal can be mostly suppressed upon CDK4/6 inhibition not only in G1 but also in S and G2 (*Figure 1—figure supplement 4*, and Materials and method section). Thus, by correcting the contribution from CDK2 activity, the reporter can be used for CDK4/6 activity measurements during the cell cycle. We therefore refer to the reporter below as a 'CDK4/6 reporter'.

## Delayed but rapid CDK4/6 activation is followed by CDK2 activation in cells exiting quiescence

When revisiting the data in *Figure 1D* and *Figure 1—figure supplement 1*, we can now conclude that mitogen stimulation triggers rapid activation of CDK4/6 but only after a 5–13 hr long delay. When comparing the kinetics of CDK4/6 activation between different cells in the same population after mitogen stimulation, 54% of cells in this experiment persistently increased CDK4/6 activity (CDK4/6$^{high}$ cells) while the remaining cells did not show an increase in CDK4/6 activity (CDK4/6$^{low}$ cells) (*Figure 2A*). The finding of a rapid increase in CDK4/6 activity after a long delay provides support for a previous hypothesis that CDK4/6 activation might be controlled by an ultrasensitive activation mechanism (*Yang et al., 2017*). Furthermore, while the activity of CDK4/6 rapidly increases to a near maximal level before S-phase, CDK4/6 activity then stays persistently high throughout S and G2 phase as evidenced by cells showing persistent CDK4/6 reporter activity after the APC/C$^{Cdh1}$ degron signal starts to increase (*Figure 2A* and *Figure 1—figure supplement 1B*, *Figure 2—figure supplement 1*). This observation is at first surprising since cyclin D levels are known to be much lower in S-phase. Nevertheless, cyclin D1 is stabilized in G1 by p21 (*Chen et al., 2013*) and p21 is degraded in S-phase by Crl4$^{Cdt2}$ (*Abbas et al., 2008*). The low levels of p21 in S-phase may then allow for maintenance of CDK4/6 activity at a lower cyclin D level, providing a potential explanation how CDK4/6 activity can stay elevated in S phase.

Previous bulk-cell analysis of cells exiting quiescence (*Baldin et al., 1993*; *Koff et al., 1992*; *Meyerson and Harlow, 1994*; *Sherr and Roberts, 1999*) suggested that CDK4/6 is activated during G1 before the activation of CDK2. To study the relative order and kinetics of CDK4/6 and CDK2 activation in single cells, we classified in *Figure 2A* mitogen-stimulated quiescent cells according to when they turn CDK4/6 activity on (as shown in *Figure 1D*, marked as CDK4/6$^{high}$ or CDK4/6$^{low}$

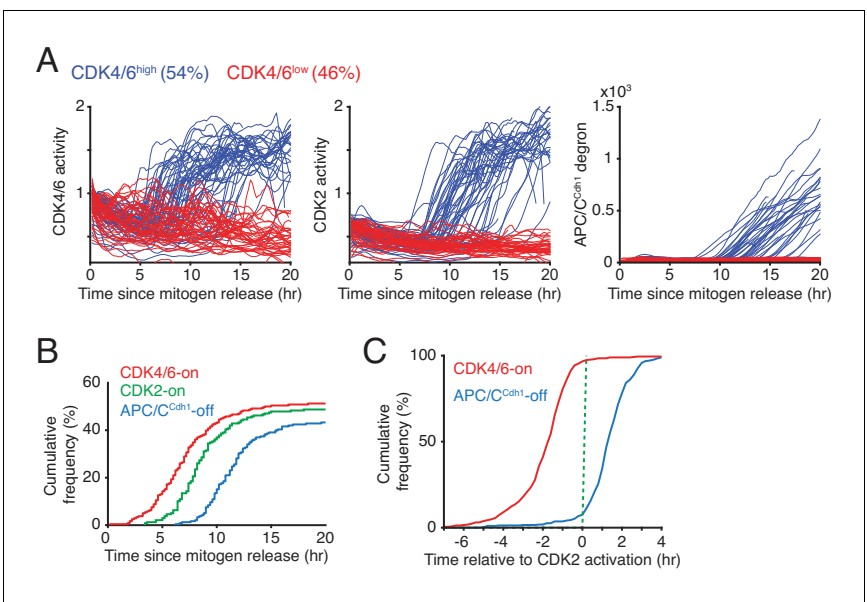

**Figure 2.** Rapid CDK4/6 activation after a long delay - followed by sequential CDK2 activation and APC/C$^{Cdh1}$ inactivation in cells exiting quiescence. (**A**) Single-cell activity traces of CDK4/6, CDK2, and APC/C$^{Cdh1}$. Cells were stimulated with mitogens after serum starvation and classified based on the detection of CDK4/6-on point (CDK4/6$^{high}$ or CDK4/6$^{low}$). (**B**) Cumulative plot of activation of CDK4/6 and CDK2, and inactivation of APC/C$^{Cdh1}$ as a function of time after mitogen release (n > 200 cells; CDK4/6-on, CDK2-on and APC/C-off points shown in *Figure 1D*). (**C**) Cumulative plots of the fraction of cells activating CDK4/6 and inactivating APC$^{Cdh1}$ as a function of time relative to CDK2 activation (n > 350 cells).

The online version of this article includes the following figure supplement(s) for figure 2:

**Figure supplement 1.** Bimodal distribution of cyclin D-CDK4/6 activity in G1 phase.

**Figure supplement 2.** p16 suppresses the fraction of cells activating CDK4/6 but does not affect the rate of CDK4/6 activation or the sequential activation kinetics of CDK2 relative to CDK4/6.

cells). The delayed rapid increase in CDK4/6 activity is almost immediately followed by an increase in CDK2 activity before APC/C$^{Cdh1}$ is inactivated after a short additional delay (*Figure 2A–C*). When we aligned single-cell time courses by the mitogen-triggered increase in CDK2 activity, we found evidence for an orderly progression whereby a rapid and persistent CDK4/6 activation occurs in nearly every cell shortly before CDK2 activation and that CDK2 activation occurs in nearly every cell before APC/C$^{Cdh1}$ inactivation (*Figure 2C*). In contrast, the CDK4/6$^{low}$ cells in the same population kept CDK2 activity low and APC/C$^{Cdh1}$ active for over 20 hr (*Figure 2A*).

One potential concern with the timing of CDK4/6 and CDK2 activation was that the immortalized MCF10A cells lack copies of the INK4 family CDK4/6 inhibitor genes p16 and p15 (CDKN2A/B) (*Cowell et al., 2005*) and may therefore have a different kinetics of CDK4/6 activation compared to cells that have expressed INK4 CDK inhibitors. To test for a potential altered kinetics in cells expressing p16, we introduced a DHFR regulated expression system into MCF10A cells to regulate the expression of exogenous p16 (*Figure 2—figure supplement 2A-B*). Even though the percentage of cells that activated CDK4/6 and CDK2 upon p16 induction was reduced, the cells that did increase CDK4/6 activity had a similar delay time between CDK4/6 to CDK2 activation (*Figure 2—figure supplement 2C-D*), and a similar rate of CDK4/6 activity increase (*Figure 2—figure supplement 2E-F*), suggesting that the observed rapid kinetics of CDK4/6 activation is not the result of cells lacking p16. Together, these experiments and controls suggest that cell-cycle entry can be a series of sequential events, with CDK4/6 activity rapidly increasing after a long and variable delay, which is immediately followed by a slower delayed increase in CDK2 activity, and which is in turn followed by rapid APC/C$^{Cdh1}$ inactivation at the onset of S phase.

## Newly born cells can have active CDK4/6 and a short G1 or initially inactive CDK4/6 and a variable longer G1

We thus far focused on CDK4/6 activity changes in cells entering the cell cycle from quiescence. We next compared the kinetics of CDK4/6, CDK2, and APC/C$^{Cdh1}$ activity changes when cells exit mitosis and have to decide whether to enter quiescence or another cell cycle. Previous studies showed that CDK2 reporter analysis can distinguish three paths in cells exiting mitosis: cells can immediately increase CDK2 activity (CDK2$^{inc}$ cells) or cells keep CDK2 activity persistently low (CDK2$^{low}$ cells) (*Spencer et al., 2013*). In a third path, cells first keep CDK2 activity low at mitosis exit and then increased CDK2 activity again after a variable delay (CDK2$^{delay}$ cells) (*Arora et al., 2017*; *Barr et al., 2017*; *Moser et al., 2018*; *Schwarz et al., 2018*; *Spencer et al., 2013*; *Yang et al., 2017*).

When we aligned asynchronously cycling cells by the end of mitosis (anaphase), a subset of cells had continuously active CDK4/6 (CDK4/6$^{high}$ cells) starting at the end of mitosis and these cells corresponded to the CDK2$^{inc}$ cells (*Figure 3A*, left). CDK4/6$^{high}$ cells were defined as having CDK4/6 activity above a threshold (0.7) for two hours after mitosis (see *Figure 3A*, right, for threshold marked by dashed red line). Another subset of cells had low CDK4/6 activity at mitotic exit and some of these cells kept CDK4/6 activity persistently off after mitosis, corresponding to CDK2$^{low}$ cells that exit to quiescence (CDK4/6$^{low}$ cells were defined as having CDK4/6 activity below threshold at 2 and 10 hr). A third subset of cells had first CDK4/6 activity below the threshold and increased CDK4/6 above the threshold typically before 10 hr after mitosis (CDK4/6$^{delay}$ cells) (*Figure 3A*, top left), corresponding to CDK2$^{delay}$ cells. Consistent with cells entering the cell cycle from quiescence, cells that persistently activate CDK4/6 also activated CDK2 and inactivated APC/C$^{Cdh1}$ when cells exit mitosis (*Figure 3A*, middle and bottom left).

A histogram analysis of CDK activities at mitotic exit showed CDK4/6 reaching a maximum activity 2 hr after mitosis, whereas CDK2 activity gradually increased (*Figure 3A*, right). Furthermore, CDK4/6 activity stays maximal when cells start DNA replication, while CDK2 activity starts elevated and then keeps gradually increasing before and after S-phase entry (*Figure 1*, *Figure 3—figure supplement 1A*). Thus, CDK4/6 activity in MCF10A cells can generally be considered at a given time in the cell cycle to be either in an 'on' or 'off' state both in cycling cells as well as in cells coming out of quiescence while CDK2 activity is gradually increasing throughout G1, S, and G2 phases.

CDK4/6$^{delay}$ cells that exit mitosis have CDK4/6 and CDK2 activity both off when they exit mitosis but then again, after a variable delay, turn both activities sequentially on to re-enter the cell cycle. The exit of cells after mitosis into a transient G0-like period and the sequential reactivation of CDK4/6 and CDK2 can be seen by selecting examples of cells with different time periods of low CDK4/6 activity after mitosis and plotting the corresponding changes in both CDK4/6 and CDK2 activity

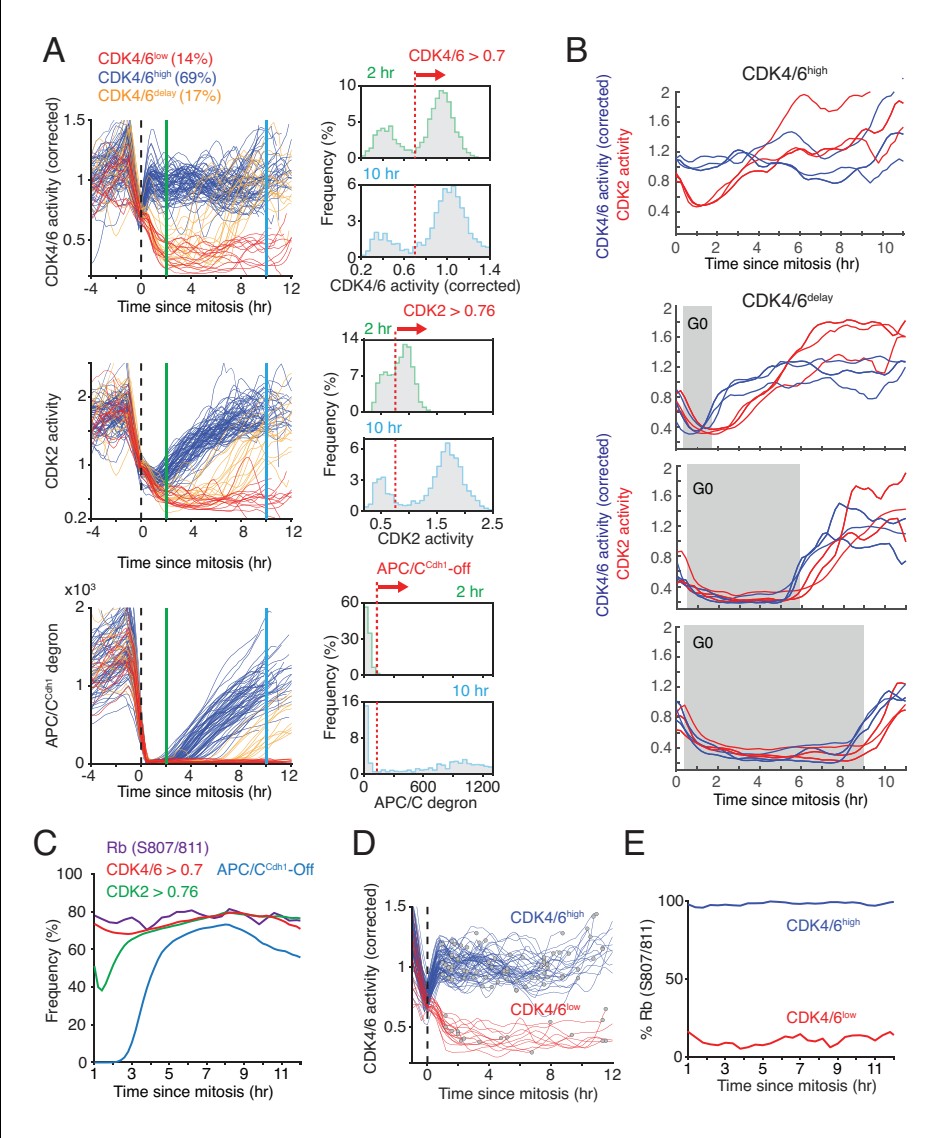

**Figure 3.** Bimodal CDK4/6 activity correlates with a bifurcation in Rb hyperphosphorylation and CDK2 activation in cells exiting mitosis or a variable period of low CDK4/6 activity after mitosis. (**A**) Left: Asynchronously cycling cells aligned by the end of the mitosis (anaphase), comparing CDK4/6, CDK2, and APC/C$^{Cdh1}$ activity changes. Corrected CDK4/6 activity signals are shown. Cells were classified by their CDK4/6 activity 2 hr after anaphase as being CDK4/6$^{high}$ or CDK4/6$^{low}$, and a subset of CDK4/6$^{low}$ cells were further classified as CDK4/6$^{delay}$ if they had initially low CDK4/6 activity but later again increased CDK4/6 activity. Right: Activity histograms show that CDK4/6 but not CDK2 activity is rapidly activated with a bimodal activity distribution 2 hr after mitosis. Dotted line indicates the threshold used to classify the subpopulation of cells with CDK4/6, CDK2 or APC/C$^{Cdh1}$ activity on or off (n > 2,000 cells). (**B**) Examples of cells that either kept CDK4/6 activity on at mitotic exit (top), or exited to an approximately 2, 6 or 9 hr long transient G0 with low CDK4/6 activity (lower three panels). Corresponding CDK2 activities are shown to highlight that both CDK activities are low during a transient G0 and are sequentially activated (3 cells shown for each transient G0 time period). (**C**) A cumulative representation of the fraction of cells with active CDK4/6 (>0.7), hyperphosphorylated Rb (see *Figure 2—figure supplement 1B*), activated CDK2 (>0.76) and inactivated APC/C$^{Cdh1}$ as a function of time after mitosis (n > 2,000 cells). (**D,E**) Live-fixed analysis of cells in the same population with active or inactive CDK4/6 at different times after anaphase. The plot in (**E**) shows the fraction of cells that have Rb phosphorylated or not in cells were CDK4/6 activity is high or low as a function of time after anaphase (n > 2,000 cells).

The online version of this article includes the following figure supplement(s) for figure 3:

**Figure supplement 1.** CDK2 and CDK4/6 activity time-courses in asynchronously cycling RPE-1 hTERT cells with and without inhibition of CDK4/6 activity by palbociclib.

time-courses in the same cells (**Figure 3B**), showing again a close coupling between CDK4/6 and CDK2 activities. In conclusion, cells that exit mitosis with active CDK4/6 and immediately increase CDK2 activity have a short G1 and immediately start the next cell cycle, while cells with inactive CDK4/6 and CDK2 have a longer variable G1 or they exit to quiescence. Thus, by varying the delay period before reactivating CDK4/6, newly born cells can regulate the length of G1.

Previous studies have shown that cycling cells that exit mitosis have a bimodal distribution in Rb hyperphosphorylation (**Cappell et al., 2016**; **Moser et al., 2018**; **Spencer et al., 2013**; **Yang et al., 2017**). We therefore tested how CDK4/6, CDK2, and APC/C$^{Cdh1}$ activities relate to the phosphorylation of Rb. As a marker of hyperphosphorylated Rb, we measured Rb phosphorylation at Serine 807/811, which shows a bimodal distribution and is correlated with activation of E2F transcription targets (**Cappell et al., 2016**; **Chung et al., 2019**; **Yang et al., 2017**). To test for a correlation between CDK4/6 activity and Rb phosphorylation, we used live-cell imaging followed by immunostaining and classified cells based on the time since mitosis. When we set a threshold for CDK4/6 and CDK2 activation, APC/C$^{Cdh1}$ inactivation (**Figure 3B**, right) and Rb phosphorylation (**Figure 3—figure supplement 1B**), we found that the fraction of cells that has active CDK4/6 and increasing CDK2 activity after mitosis closely matches the fraction of cells with phosphorylated Rb (**Figure 3C**), suggesting that activation of CDK4/6 is closely correlated with Rb phosphorylation. We next classified cells into CDK4/6$^{high}$ and CDK4/6$^{low}$ populations (using the threshold from **Figure 3A**) and determined whether Rb is phosphorylated in these two populations at different times after mitosis. This analysis confirmed that the CDK4/6$^{high}$ versus CDK4/6$^{low}$ population remains closely correlated over time with whether Rb is phosphorylated for the measured 12 hr period after mitosis (**Figure 3D,E**).

The percent of cells in a population that re-enter or exit the cell cycle after mitosis, or have a delayed cell-cycle entry path with variable G1, were previously shown to vary between different cell types and mitogen and stress stimulation conditions in culture (**Cappell et al., 2016**; **Moser et al., 2018**; **Schwarz et al., 2018**; **Spencer et al., 2013**). To test whether the bimodal characteristic of CDK4/6 after mitosis is unique to MCF10A cells, we also measured CDK4/6 activity changes at mitotic exit in another commonly used immortalized retinal pigment epithelial cell model, RPE1 cells. An analysis of CDK4/6 and CDK2 activity changes after mitosis in RPE1 cells showed a similar bimodal behavior as in MCF10A cells with cells either having active CDK4/6 followed by CDK2 activation, or inactive CDK4/6 without that CDK2 is activated (**Figure 3—figure supplement 1A**). Treatment with the CDK4/6 inhibitor palbociclib also abolished both CDK4/6 and CDK2 reporter signals after mitosis (**Figure 3—figure supplement 1B**).

Taken together, our data suggest that cells with high CDK4/6 activity and increasing CDK2 activity at mitotic exit also have Rb hyperphosphorylated which in turn directs cells to rapidly re-enter the cell cycle after a short G1. Alternatively, cells with low CDK4/6 activity at mitotic exit keep CDK2 activity off and Rb dephosphorylated and exit to quiescence, or alternatively, cells can again rapidly increase CDK4/6 activity after a variable delay of low CDK4/6 activity and enter the next cell cycle, thereby generating G1 lengths of variable duration.

## Stress can rapidly inactivate CDK4/6 but with reduced efficacy as cells increase CDK2 activity towards the end of G1

Previous studies showed that cell-cycle entry can be reversed in G1 by different types of stresses even after CDK2 starts to increase - as long as cells have not yet inactivated APC/C$^{Cdh1}$ (**Cappell et al., 2016**; **Heldt et al., 2018**). This raises the questions whether or when different stresses can suppress CDK4/6 or CDK2 activity to reverse cell-cycle entry. To activate two different stress pathways, we used $H_2O_2$ to increase reactive oxygen signaling and tunicamycin to trigger an ER stress response. To investigate how CDK4/6 and CDK2 activities respond to these stresses, we stimulated serum-starved cells with mitogens, applied $H_2O_2$ and tunicamycin at 11 hr after mitogen release, and selected cells where CDK4/6 and CDK2 activity have previously increased and cells were in G1 or S phase according to the APC/C$^{Cdh1}$ reporter signal. We classified cells by whether they continued along in the cell cycle or reversed cell-cycle entry as evidenced by falling CDK2 activity (CDK2$^{inc}$ versus CDK2$^{low}$). For both types of stresses, a subset of cells in G1 inactivated CDK4/6 activity and reversed cell-cycle entry (**Figure 4A**).

Intriguingly, the outcome whether cells will inactivate CDK4/6 and exit the cell cycle was closely correlated with the level of CDK2 activity when stress signals were applied (**Figure 4B**). Cells having lower CDK2 activity were more likely to inactivate CDK4/6 and exit to quiescence after addition of

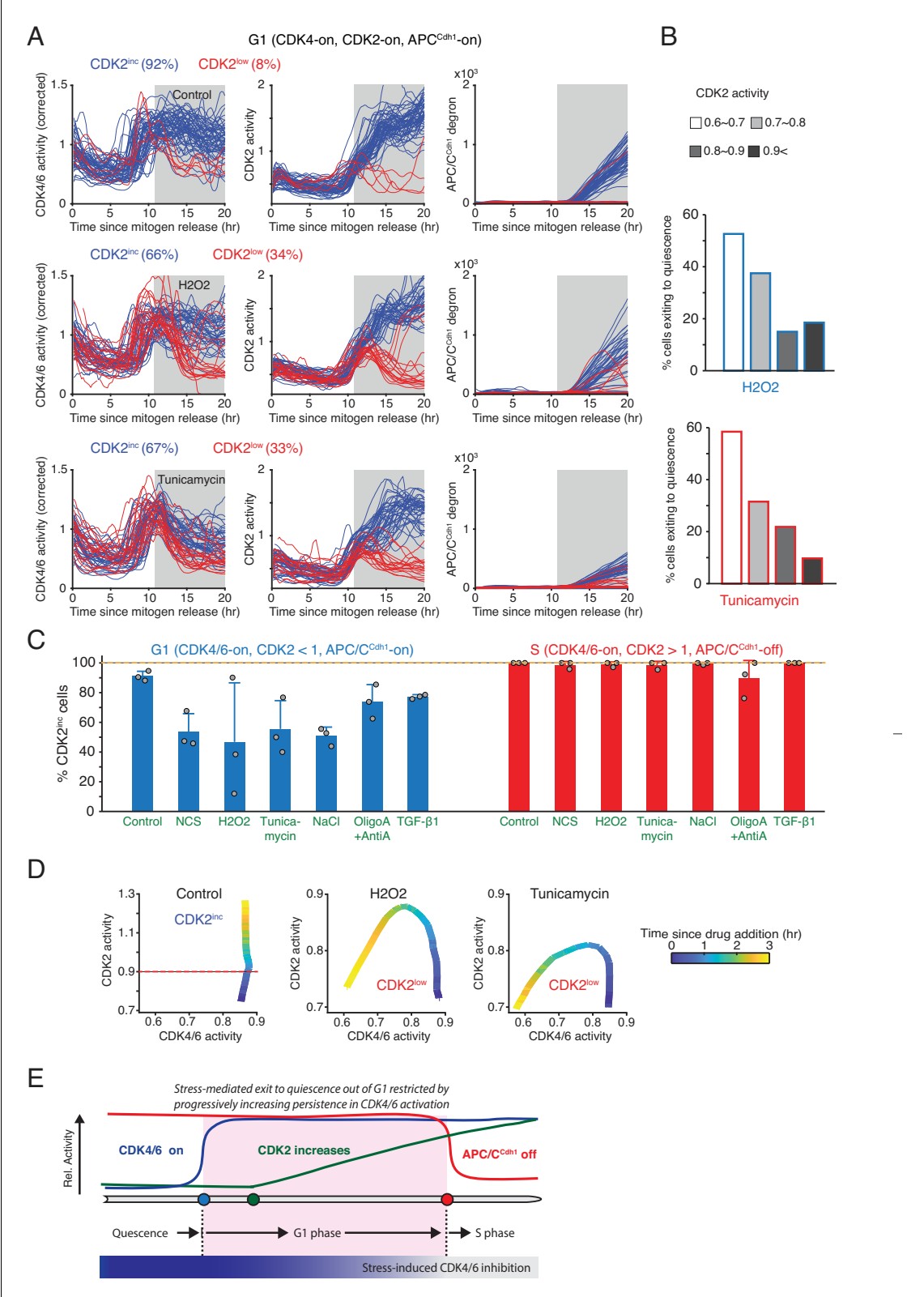

**Figure 4.** Increased resistance against stress-mediated CDK4/6 inactivation as cells increase CDK2 activity towards the end of G1 phase. (**A**) Activity changes of the two CDK reporters and APC/C$^{Cdh1}$ in G1 phase in response to stress: H2O2 (200 μM) and Tunicamycin (5 μM). Cells were classified based on the resulting CDK2 activity changes after addition of stresses (CDK2$^{inc}$ or CDK2$^{low}$, measured at 15 hr). Cells included in the analysis were selected if they had active CDK4/6 and CDK2 activity and no APC/C degron signal at 11 hr after mitogen release when the stresses were applied (gray

*Figure 4 continued on next page*

*Figure 4 continued*

area). (B) % cells inactivating CDK2 activity in response to H2O2 and tunicamycin. Cells were classified based on CDK2 activity when stresses were added (n > 20 cells from 3 independent experiments). (C) % of CDK2$^{inc}$ cells in response to various types of stresses. Data are mean ± s.d. (3 independent experiments). (D) Phase plot analysis of CDK2 versus CDK4/6 activity after application of different stresses. Time is color-coded (n > 35 cells from 3 independent experiments). (E) Schematic representation of the progressive loss in stress sensitivity of CDK4/6 activity as cells approach the onset of S phase.

The online version of this article includes the following figure supplement(s) for figure 4:

**Figure supplement 1.** Control experiments showing that cells become resistant to stress in S phase.
**Figure supplement 2.** Probability of cells reversing CDK4/6 activation and cell-cycle entry decreases with increasing CDK2 activity.

stress signals while the probability became progressively lower as CDK2 activity increased. Since CDK2 activity gradually increases towards the start of S phase, the increasing CDK2 activity can be seen as a timer, which shows that CDK4/6 activity becomes less sensitive to stress as cells approach S phase. In contrast, there was no correlation between the amplitude of the acute CDK4/6 reporter signal and cell-cycle exit, likely since the CDK4/6 reporter signals were high but had typically plateaued out before stress application (*Figure 4—figure supplement 1A*). Consistent with a correlation between G1 and S progression and stress-mediated reversal, nearly all cells that already entered S phase and had high CDK2 activity (CDK2 >1.0) did not respond to H$_2$O$_2$ or tunicamycin (*Figure 4C* and *Figure 4—figure supplement 1B*). Thus, as cells progress in G1 towards S phase, the probability for an individual cell to inactivate CDK4/6 and CDK2 and exit the cell cycle in response to stress progressively decreases until the G1/S transition and then stays low during S phase.

When comparing the relative kinetics of CDK4/6 and CDK2 inactivation, both H$_2$O$_2$ and tunicamycin inactivated CDK4/6 before CDK2 activity started to decrease (*Figure 4D*). This difference in kinetics can best be seen in a phase plot analysis where time is color coded and CDK2 activity is plotted as a function of CDK4/6 activity. Notably, H$_2$O$_2$ inactivated CDK4/6 activity only in CDK2$^{low}$ cells, whereas tunicamycin inactivated CDK4/6 both in CDK2$^{low}$ and CDK2$^{inc}$ cells (*Figure 4A,D* and *Figure 4—figure supplement 1C*).

Finally, we tested how general these CDK4/6 inactivation responses are by using as additional stresses a pulse of Neocarzinostatin (NCS) to generate double strand DNA breaks, high salt (NaCl) to trigger osmotic stress, and oligomycin A and antimycin A to reduce mitochondria function and ATP production, and we also activated the TGF-β1 signaling pathway, which antagonizes cell proliferation (*Fräter-Schröder et al., 1986*). Similar as H$_2$O$_2$ and tunicamycin, all stress and TGF-β1 signaling were more likely to reverse cell-cycle entry in cells having lower CDK2 activity (*Figure 4C* and *Figure 4—figure supplement 2A, B*). While NaCl rapidly suppressed CDK4/6 and CDK2 activity at a similar rate, all other stresses and TGF-β1 signaling first inactivated CDK4/6 before CDK2 (*Figure 4—figure supplement 2C*). The typically more rapid stress-mediated inactivation of CDK4/6 is likely functionally relevant for the subsequent CDK2 inactivation as CDK4/6 activity is still needed to sustain the CDK2 activity increase throughout G1 phase (*Figure 1D*) until Rb hyperphosphorylation becomes independent of CDK4/6 activity at the onset of S phase (*Chung et al., 2019*).

Together, our results are consistent with the interpretation that stress and inhibitory signals generally act by first inhibiting CDK4/6 activity which in turn suppresses CDK2 activity, arguing that CDK4/6 is in G0/G1 not only rapidly regulated by mitogens but also by stress. Importantly, there is a progressive persistence in CDK4/6 activity in response to stress as cells approach the G1/S transition.

## Discussion

Our study reports the design of a CDK4/6 activity reporter based on a cyclin D-CDK4/6-specific kinase binding site derived from Rb (*Topacio et al., 2019*; *Wallace and Ball, 2004*), and a modular strategy to develop protein kinase reporters (*Regot et al., 2014*). We used immortalized MCF10A breast epithelial cells to engineer a reporter system to monitor in the same cell activity changes of CDK4/6 and CDK2, as well as of the E3 ubiquitin ligase APC/C$^{Cdh1}$ that is inactivated at the end of G1 phase. We show that the CDK4/6 reporter is best used as a dual reporter together with a CDK2 reporter to subtract a CDK2 contribution from the CDK4/6 reporter signal. The corrected CDK4/6

reporter signal stays elevated in mitogen-stimulated cells after cells enter S phase, and can be inhibited during S and G2 by CDK4/6-selective inhibitors, suggesting that cells activate CDK4/6 in G1 and can then keep CDK4/6 active throughout the cell cycle.

Using this reporter system, we first explored the kinetics of CDK4/6 activation. A previous study by our group calibrated nuclear cyclin D1 and p21 expression levels and showed that only cells with at least a small excess of nuclear cyclin D1 over p21 after mitosis have Rb hyperphosphorylated while cells with more p21 than cyclin D1 had Rb invariably dephosphorylated (*Yang et al., 2017*). One interpretation of this result was that a competition between cyclin D1 and p21 makes it possible for a gradual increase in the relative level of cyclin D1 versus p21 to trigger a rapid activation of CDK4/6 once cyclin D1 exceeds the level of p21. We considered that such a rapid CDK4/6 activation may explain the observed bimodal hyperphosphorylation of Rb. In support of a hypothesis that CDK4/6 can be activated in an ultrasensitive, bimodal manner, we found that the CDK4/6 reporter signal typically increased rapidly after an often very long variable delay both in cells exiting quiescence and in cells exiting from a transient period of low CDK4/6 activity after mitosis.

Previous studies of MCF10A and other cell types further showed that cells in the same population have Rb typically either hyperphosphorylated or dephosphorylated after mitosis with the percent of cells having Rb hyperphosphorylated being regulated by mitogen and stress signaling and also differ between cell types (*Cappell et al., 2016*; *Moser et al., 2018*; *Schwarz et al., 2018*; *Spencer et al., 2013*; *Yang et al., 2017*). In cells with hyperphosphorylated Rb at mitotic exit, the activity of CDK2 was shown to immediately increase and cells entered the next S phase a few hours later, while cells with low Rb phosphorylation kept their CDK2 activity low after mitosis and cells had either a variable G1 length or they exited to quiescence. Our study here shows that the bimodal hyperphosphorylation of Rb and bimodal CDK2 activation can be explained by an underlying bimodality in CDK4/6 activity at mitotic exit.

Together with previous results, our measurements with the CDK4/6 reporter support the hypothesis that cells choose between three paths when they exit mitosis: In one subset of cells, cells keep CDK4/6 active and Rb hyperphosphorylated as cells exit anaphase and mitosis and these cells are characterized by an immediate increase in CDK2 activity and a short G1 length. A second subset of cells have CDK4/6 activity low and Rb dephosphorylated at mitotic exit and CDK4/6 and CDK2 activity then stay persistently low until cells enter quiescence or a postmitotic state. A third subset of cells start the same way with CDK4/6 and CDK2 being inactive and Rb dephosphorylated after mitosis, but after a variable delay, these cells will again increase CDK4/6 activity, hyperphosphorylate Rb, and increase CDK2 activity to re-enter the next cell cycle. Several previous studies have shown that G1 is the most variable phase of the cell cycle and regulation of G1 length is important to regulate cell differentiation (*Lange and Calegari, 2010*). Our study thus suggests that cells can control G1 length and cell-cycle duration in part by switching CDK4/6 activity off for a variable time period after mitosis.

Our study further shows that CDK4/6 and CDK2 activity become progressively more resistant to stress as cells approach S phase. While previous studies showed that stress can trigger cell-cycle exit in G1 until the onset of S phase (*Cappell et al., 2016*), and that stress-mediated inactivation of CDK2 becomes less effective towards the onset of S phase (*Heldt et al., 2018*), if and how stress alters CDK4/6 signaling was not yet understood. We measured the response to a number of different types of stress signals and showed that most trigger first a rapid inactivation of CDK4/6 and that the loss in CDK4/6 activity is then followed by a reduction in CDK2 activity and exit to quiescence. This is also consistent with CDK2 activity and cell-cycle entry remaining sensitive to acute inhibition by the CDK4/6 inhibitor palbociclib until the end of G1 (*Chung et al., 2019*). We further show that CDK4/6 activity becoming increasingly more resistant to stress towards the end of G1. Such a hysteresis in CDK4/6 activity in response to stress was unexpected since the switch-mechanism driving cell-cycle entry is typically believed to self-reinforce Rb hyperphosphorylation and CDK2 activity.

In conclusion, our study reports the development of a reporter system to monitor CDK4/6 and CDK2 activities live in single cells as a method to help answer remaining questions about how mitogens, contact inhibition, and stress inputs control CDK4/6 and CDK2 activation and the cell-cycle entry decision. In a first application of the reporter system, we here show (i) that CDK4/6 activity can change rapidly when cells exit quiescence or mitosis, (ii) that temporary inactivation of CDK4/6 after mitosis regulates G1 length, (iii) that CDK4/6 activity is closely correlated with subsequent Rb hyperphosphorylation and CDK2 activation, and (iv) that stress can rapidly inactivate CDK4/6 in G1 but

progressively less so as cells approach S phase. Our study nevertheless left many questions open about the molecular mechanisms how CDK4/6 is activated and the role of CDK4/6 activity in regulating cell-cycle entry. Since we mostly used MCF10A cells, further studies will also be needed to evaluate how general our observations are and how CDK4/6 activity is regulated in different cell types and for different experimental conditions. There are for example well-known differences in the regulation of CDK4/6 and CDK2 activities between normal and cancer cells, and our study may serve as a roadmap how simultaneous CDK4/6 and CDK2 reporter analysis can be applied to understand therapeutically relevant signaling differences between normal and cancer cells.

## Materials and methods

### Cell culture

MCF10A human mammary epithelial cells (ATCC, CRL-10317) were cultured in phenol red-free DMEM/F12 (Invitrogen) supplemented with 5% horse serum, 20 µg/ml EGF, 10 µg/ml insulin, 0.5 µg/ml hydrocortisone, 100 ng/ml cholera toxin, 50 U/ml penicillin, and 50 µg/ml streptomycin. For serum starvation, the growth medium without horse serum, EGF, and insulin was used. All cell lines tested negative for mycoplasma.

RPE1-hTERT retinal epithelial cells (ATCC CRL-4000) were cultured in phenol red-free DMEM/F12 supplemented with 10% FBS. For experiments performed at lower concentrations of FBS, cells were first plated and several hours later transferred into media containing the indicated FBS concentration.

### Antibodies and reagents

Palbociclib, Abemaciclib, Ribociclib, and Roscovatin were from Selleckchem. CDK2i III was from Millipore. NaCl, Neocarzinostatin, and hydrogen peroxide were from Sigma-Aldrich. Tunicamycin was from Cayman Chemical. Oligomycin A and Antimycin A were from Abcam, TGF-β1 and Rabbit anti-phospho-Rb (Ser807/811) (#8516) were from Cell Signaling Technology.

### Constructs and stable cell lines

The live-cell CDK4/6 reporter was cloned into a pLenti vector using the Gibson cloning method to assemble fragments of mCherry, a peptide containing a bipartite nuclear localization sequence (bNLS) and a nuclear export sequence (NES), and the C-terminus of Rb containing amino acids 886–928 (pLenti-mCherry-CDK4KTR). Using Gibson cloning, DHB-mVenus-p2a-mCherry-CDK4KTR and mCerulean-Geminin (aa1-110)-p2a-H2B-iRFP were cloned into pLenti vector. To generate stable cell lines, lentiviral constructs were introduced into MCF10A cells by viral transduction. As added notes, we had difficulties constructing responsive CDK4/6 reporters when replacing mCherry with different fluorescent proteins. This may suggest that mCherry conjugating contributes to an optimal performance of the CDK4/6-regulated import-export regulation. Also, due to higher stability of mCherry in low pH compared to other reporters, the CDK4/6 reporter can accumulate in lysosomes for some experimental conditions, which can add a background cytoplasmic signal in the ratio-analysis.

### Immunofluorescence

Cells were fixed by adding 4% paraformaldehyde at a ratio of 1:1 to culture medium (final 2% paraformaldehyde) for 15 min. Then, cells were washed three times in PBS, followed by incubation in permeabilization/blocking with 0.1% triton X-100, 10% FBS, 1% BSA, and 0.01% NaN$_3$ for 1 hr, and stained overnight at 4°C with primary antibodies. Primary antibodies were visualized using a secondary antibody conjugated to Alexa Fluor-488, Alexa Fluor-568, and Alexa Fluor-647. For EdU staining, cells were treated with 10 µM EdU for 15 min and fixed and processed according to manufacturer's instructions (Invitrogen, #C10356).

### CDK4/6 activity correction based on the activity of the CDK2 reporter

To calculate the contribution of CDK2 activity to the CDK4/6 reporter in S/G2 phase, we used CDK4/6 inhibition and linear regression. This analysis showed that multiplication of a correction factor of approximately 0.35 to the measured CDK2 activity results in a near constant derived reporter

signal during the cell cycle after CDK4/6 inhibition. We term the resulting signal a 'corrected CDK4/6 activity'.

Corrected CDK4/6 activity = CDK4/6 reporter activity - 0.35*CDK2 reporter activity.

## Microscopy

All Images were taken using 20X objective (0.75 N.A.) on a IXmicro microscope (Molecular Decies). Cells were imaged every 12 min in a 37°C chamber at 5% $CO_2$.

## Image analysis

Segmentation and tracking performed as previously described *Cappell et al. (2016)*; *Yang et al. (2017)*.

## Acknowledgements

We thank Yan Geng and Pior Sicinski for the E/A-null MEF cell line. This work was supported by a Seed Grant awarded to HY, SC, AJ, and SR by the Stanford Center for Systems Biology (P50 GM107615) and R35 GM12702602 (TM). SDC was supported by a Damon Runyon Cancer Research Foundation Fellowship (DRG-2141). MC is supported by the Paul. G Allen Discovery Center at Stanford University.

## Additional information

### Funding

| Funder | Grant reference number | Author |
| --- | --- | --- |
| National Institute of General Medical Sciences | GM127026 | Tobias Meyer |
| Stanford Center for Systems Biology | P50 GM107615 | Hee Won Yang<br>Steven D Cappell<br>Ariel Jaimovich<br>Sergi Regot |
| Paul. G Allen Discovery Center | | Markus Covert |

The funders had no role in study design, data collection and interpretation, or the decision to submit the work for publication.

### Author contributions

Hee Won Yang, Conceptualization, Data curation, Formal analysis, Validation, Investigation, Methodology; Steven D Cappell, Ariel Jaimovich, Conceptualization, Investigation, Methodology; Chad Liu, Investigation; Mingyu Chung, Methodology; Leighton H Daigh, Lindsey R Pack, Yilin Fan, Investigation, Methodology; Sergi Regot, Conceptualization; Markus Covert, Conceptualization, Funding acquisition; Tobias Meyer, Conceptualization, Supervision, Funding acquisition, Validation, Project administration

### Author ORCIDs

Sergi Regot http://orcid.org/0000-0001-9786-3897
Markus Covert http://orcid.org/0000-0002-5993-8912
Tobias Meyer https://orcid.org/0000-0003-4339-3804

### Decision letter and Author response

Decision letter https://doi.org/10.7554/eLife.44571.sa1
Author response https://doi.org/10.7554/eLife.44571.sa2

## Additional files

### Supplementary files

- Source data 1. Original data used for analysis in different figure panels figures.

- Transparent reporting form

### Data availability

The source data of the different time course analysis is available as a ZIP file (Source data 1).

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
