## [Decision Letter]

**Acceptance summary:**

In this study, Yang and colleagues (Meyer lab) developed a novel fluorescent reporter that allows live tracking of CDK4/6 kinase activity in growing cell populations. Cyclin D-CDK4/6 kinases are key players in cell cycle-entry control, which is almost universally deregulated in human cancer. Hence, a reporter for their activity will be valuable, and the authors provide a convincing case that the novel reporter can be used to accurately follow Cyclin D-CDK4/6 activity in living cells. By incorporating the CDK4/6 activity sensor into a previously developed strategy, one can now visualize G1 progression in individual cells, based on the automatic tracking of live-cell reporters for three critical cell cycle regulators. Following this strategy, the authors show that CDK4/6 kinases are rapidly activated after a variable lag time, which depends on the individual cell, consistent with an ultrasensitive activation mechanism. This activation apparently precedes pRb-Ser807/811 phosphorylation, cyclin E-CDK2 activation, and APC/C-Cdh1 inactivation. As a major novel finding, various stress signals are found to downregulate CDK4/6 kinase activity and to trigger cell cycle arrest until the point when the activity of cyclin E-CDK2 kinases has reached a critical level, near the G1/S transition. Thus, the combined use of three different live-cell reporters has allowed the authors to track the cell cycle behavior in hundreds of individual cells, which confirmed previously established relationships between cell cycle regulators and revealed novel aspects of cell cycle entry control.

This revised manuscript includes an important additional experiment in which DHFR-p16 is introduced to re-express the p16 inhibitor in MCF10a cells. Although this reduces the fraction of cells that activate CDK4/6, the kinase still shows the same abrupt activation in the presence of p16. These results and the previously added cyclin E and A knockout experiments provide strong additional support for the conclusions of the study. Moreover, the manuscript has been substantially revised, with more careful interpretation of the results, and now includes a more general introduction. This time, the authors very clearly state what the limitations are of their reporters and better define the term "CDK4/6 activity reporter" and "CDK2 reporter". Together, this study is of high quality and broad importance, and I recommend publication in eLife.

**Decision letter after peer review:**

Thank you for submitting your article "Stress-induced inactivation of CDK4/6 and exit from the cell cycle is progressively blocked as cells approach S-phase" for consideration by *eLife*. Your article has been reviewed by three peer reviewers, including Sander van den Heuvel as the Reviewing Editor and Reviewer #1, and the evaluation has been overseen by Anna Akhmanova as the Senior Editor. The following individual involved in review of your submission has agreed to reveal their identity: Hein te Riele (Reviewer #3). One of the reviewers has opted to remain anonymous.

The reviewers have extensively discussed the reviews with one another and the Reviewing Editor has drafted this decision to help you prepare a revised submission.

Summary:

In this study by Yang et al., the Meyer lab expands on their strategy of examining cell cycle entry with single cell resolution based on automated tracking of fluorescent CDK-reporter localization. In previous studies, this group developed and applied reporters to monitor CDK2-cyclin E/A and APC/C-Cdh1 activities. Here, yet another fluorescent reporter is created, which provides a read out for the activity of CDK4/6 in association with cyclin D. Using a variety of CDK-inhibitors, the authors provide evidence that this new reporter, which includes a previously reported Rb-derived docking site for Cdk4, indeed reflects CDK4/6 kinase activity. Using simultaneous recording of the three reporters, the authors follow the behavior of individual cells during exit from mitosis into either the next cell cycle or into a period of temporal or prolonged quiescence. The acquired time series of CDK4/6-cyclin D, CDK2-cyclin E/A, and APC/C-Cdh1 activities allows correlating the history of individual cells with their later behavior and response to e.g. drugs or stress factors. This is a powerful strategy that is used to demonstrate the temporal order of CDK4/6 activation, CDK2 activation, and APC/C-Cdh1 inactivation. Moreover, the system is used to demonstrate that cells can arrest in response to stress signals, depending on the level of CDK2 activity in the G1 phase. Cells with high CDK2-cyclin E/A kinase activity fail to respond to various stresses, whereas cells with low CDK2 kinase return to quiescence and reduce CDK activities. Moreover, the authors show that in this stress response, CDK4/6 activity declines before CDK2.

This study includes several strong points, but also some weaker aspects:

– The quality of the work is high and the apparent specificity of the reporter for CDK4/6 is quite surprising. However, the evidence supporting and explaining this specificity is currently not extensive.

– While individual cell tracing is certainly valuable, in this case the conclusions about the temporal order largely confirm the conclusions from previous studies that were based on kinase assays, Western blots and IPs from bulk cell populations.

– Insight in the kinetics of CDK4/6 activation is important and novel. A limitation here is the sole use of the MCF10a cell line, which lacks the most important CDK4/6 antagonist, p16.

– The stress response data is new and interesting. Here, one would have liked to see DNA damage included in the analysis, in particular because it remains unclear to what extent the experimental procedure – time-lapse fluorescence microscopy – induces a "stress" response that contributes to the observed bifurcation in the control cell population.

Essential revisions:

1) The development of a specific CDK4/6 sensor is of great importance, for this study as well as the field of cell cycle and cancer research. The specificity of the reporter should therefore be confirmed beyond any doubt. Currently the validation of the specificity relies entirely on the response to chemical CDK inhibitors. The high specificity of the CDK4/6 inhibitors and the time of addition (after kinase activation) puts some confidence in these experiments. However, the contribution of CDK2 in phosphorylating the CDK4/6 sensor remains a bit uncertain (see points below). Moreover, although a docking site for CDK4/6 has been added, it is unclear why this sensor would not be seen by CDK2-cyclin E/A kinases. Additional experiments are needed to validate and clarify the specificity. First and foremost, it is critical to compare and contrast the combined sensors in CDK4/6 double knockout cells. Further, the protein sequence of the reporter should be provided, and testing additional drugs, in particular against MAP kinases, mutating the Rb binding site, and performing in vitro kinase assays would help characterize and clarify the specificity of the reporter.

2) The Materials and methods section is much too condensed and needs to be expanded by describing every experiment in detail. The specific parameters that are used in the image analyses and for the statistical analyses need to be stated. The exact nucleotide sequence of the CDK4/6 reporter needs to be described.

3) Although the exact timing of reporter activity is highly variable, results clearly show in every single cell that mitogen stimulation first induces CDK4/6 activity, followed on average 2 h later by Cyclin E-CDK activity and after another 3-4h loss of APC/C-CDH1 activity which demarcates S-phase entry. The specificity of the reporters was confirmed by using several CDK inhibitors. Figure 1F shows that CDK2 inhibitors did reduce the activity of the CDK4/6 reporter, but the authors consider this "a negligible effect". What is the explanation? Non-specificity of the CDK2 inhibitors, partial response of the CDK4/6 reporter to CDK2 activity, feedback of CDK2 activity on CDK4/6 activity?

4) The next paragraph starts with: "The result of adding CDK4/6 inhibitors was different in S-phase where we found an incomplete suppression of the putative CDK4/6 reporter signal." Please make clear which figures need to be compared to arrive at this conclusion (probably the first and third panel of row 2 in Figure 1—figure supplement 2).

5) Related to this: is the conclusion that the CDK4/6 reporter partially responds to Cyclin E-CDK activity not also supported by the first panel of the second row of Figure 1—figure supplement 2? And if so, wouldn't this also be the case in G1 cells as shown in Figure 2F first panel, second row, requiring some correction in G1 cells as well?

6) The concept of "bimodal distribution" of CDK4/6 activity after mitotic exit is questionable. After all, 17% of the cells show a delayed increase in CDK4/6 activity which is not seen in the histograms of Figure 2A as this increase happens just between the chosen time points 2 and 9 h after mitotic exit.

The interesting observation is that in the presence of mitogenic stimuli, 69% of cells maintain high CDK4/6 activity whereas cyclin E/A-CDK activity invariably declines after mitotic exit. But which mechanism governs the maintenance or degradation of CDK4/6 activity right after mitosis? Is this controlled by a similar stress response as reported later in the manuscript, and could the frequent exposure to UV light, or other culture conditions, trigger this stress response in part of the control population? It would be very helpful if the authors can clarify this issue.

7) Apparently, 14% of cells exiting mitosis do not reactivate CDK4/6 (Figure 2A). What is the fate of these cells? It would be interesting to know whether these cells will die or remain quiescent.

8) Interestingly, the response of cells to several stressors depended on the level of Cyclin E/A-CDK activity: if low, cells responded by lowering CDK4/6 activity and after an initial increase also Cyclin E/A-CDK activity. In part, these different responses appear to reside in different phases of kinase activation: CDK4/6 is at max levels at the time of stress induction, while CDK2-associated kinase activity is rapidly increasing. This should be discussed. Moreover, it is not so clear how in Figure 3A a distinction is made between high and low cyclin E/A-CDK activity. Was that based on the 20 h time point? Similarly, for Figure 4—figure supplement 1: how are CDK4/6 high and low defined?

9) The definitions are important to know in order to appreciate Figure 4—figure supplement 1A and 2A showing that rather than the level of CDK4/6 activity the activity of cyclin E/A-CDK determines whether cells can arrest in response to stress. However, isn't this strange as Figure 1 shows that increasing cyclin E/A-CDK activity is preceded by increasing CDK4/6 activity? After all, the authors conclude that the probability of cells to inactivate CDK4/6 and Cyclin E/A-CDK becomes progressively smaller as the cells progress through G1 phase and approach the G1/S transition. This stage is characterized by high activity of both CDK4/6 and Cyclin E/A-CDK; so why no relation with CDK4/6 activity?

10) The panel indication in Figure 1—figure supplement 2 should preferably be adapted to match the legend.

11) The use of references is unnecessarily incomplete, some references listed lack important information.

12) The Discussion should be expanded and include a description of how the current observations relate to previously published data.

[Editors' note: further revisions were suggested prior to acceptance, as described below.]

Thank you for re-submitting your article "Stress-mediated exit to quiescence restricted by progressively increasing persistence in CDK4/6 activation" for consideration by *eLife*. Your article has been reviewed by one peer reviewer and the evaluation has been overseen by Anna Akhmanova as the Senior Editor.

The Reviewing Editor has drafted this decision to help you prepare a revised submission.

The experimental part of the work and observations are considered worth publication in *eLife*. However, the results should be interpreted with more caution, and clearly take into account that the cell line used lacks an important CDK4/6 inhibitor. Moreover, the writing should be broadened, some terminology used more carefully, and the Discussion should critically discuss what can be concluded and what cannot be learned from the current study. Further, more extensive and proper citations of others will be essential. Please see the review below for additional details.

Reviewer #1:

In this revised manuscript, Yang et al. (Tobias Meyer lab) use live-cell reporters to examine the regulation of cell cycle entry. The authors developed a novel fluorescent reporter, to allow live tracking of CDK4/6 kinase activity. As Cyclin D-CDK4/6 kinases are key regulators of cell cycle entry, the possibility to directly follow their activity is attractive and important. By incorporating the new CDK4/6 sensor into the approach previously developed by the Meyer group, it now becomes possible to follow cell cycle progression based on the automatic tracking of individual cells that express three distinct reporters with sequential activities. As a major novel finding, the strategy resulted in the detection of a stress response in G1, which counteracts CDK4/6 activity and triggers cell cycle arrest when activated before cells reach a critical level of CDK2 activity near the G1/S transition.

Several of the points raised by the reviewers have been addressed in the revised manuscript and a reorganization of some of the figures makes the flow more logical and transparent. A main concern from the reviewers was the specificity of the novel reporter for CDK4/6 kinases. The authors state more clearly now that CDK2 can contribute to reporter phosphorylation. Combined with the temporal resolution of the experiments and CDK2 (or: cyclin E/A kinase) activity curves, the corrected reporter activity appears informative for CDK4/6 activity throughout the cell cycle. Further, in response to the reviewer's questions, additional kinase inhibitors have also been tested for their effect on reporter localization. Moreover, an examination of cyclin E and A quadruple knockout cells is now included, which excludes the possibility that phosphorylation of the reporter by both CDK4/6 and CDK2 kinases would be needed to induce nuclear export.

Technically, this is a strong study with several important novel findings and observations. Its weakness is a general oversimplification and lack of critical evaluation throughout the text.

The explicit request from the reviewers to expand the Discussion and improve the citation of other work has mostly been met with increased self-citations. The manuscript, discussion, and model (Figure 4E) give the impression that the measured activity profiles provide an all-encompassing explanation for how cell cycle entry is regulated. However, as indicated in the previous review, all experiments are performed in MCF-10A cells that lack p16, the most important CDK4/6 antagonist. In fact, MCF-10A cells contain a deletion of the CDKN2A/2B locus (p16, ARF, p15), as well as a MYC amplification (Cowal et al., 2005; Worsham et al., 2006; Kadota et al., 2010). This does not necessarily disqualify the current study – the novel reporter and stress response remain important. However, when considering the CDK4/6 activation profiles and ultrasensitivity of the response, p16 and its absence in the cells studied should be a major factor to consider.

In general, critical evaluations and limitations of the experiments should be presented in a transparent way. Contradictions to the previous work, such as the study form Schwartz (Skotheim group), should be fairly cited and not just to draw a parallel with the current work.

Less dramatic, but still absolutely requiring adjustments: the Introduction still contains a fair number of questionable citations (e.g. "Yang et al., 2017" for a general and long established cell cycle regulation mechanism), while other citations are missing (e.g. "the dynamics of CDK4/6……E-CDK2 activation has not been explored"; in fact, the two kinase activities were compared as early as 1994; Meyerson et al. MCB). The inhibitors used should be considered more critically: RO-3306 is called a CDK1 inhibitor and Roscovitin a CDK2 inhibitor, but the differences in affinity are modest and at the concentrations used (60 microM) several other CDKs (including CDK2/A for RO-3306 and CDK1/A,B for Roscovitin) would be expected to be blocked. Also, the inhibitor data in Figure 1—figure supplement 2C would be worth discussing in more detail. Is sequential cyclin E-CDK2, cyclin A-CDK2, cyclin A-CDK1 kinase activity likely responsible for the prolonged increase in CDK2 reporter activation? How does the new insights fit with the recent data that challenge the traditional model for CDK4/6 function (Dyson lab 2019; Dowdi lab 2014).

There is a discrepancy in the correction factor for the CDK4/6 reporter. The text and cover letter mentions subtracting 35%, while the legend of Figure 1—figure supplement 4 states 25%, and the Materials and methods uses a factor 0.25x.

Together, the current description of the work is too one-sided, and the interpretation of the results is certainly incomplete and possibly incorrect without considering the absence of p16.

---

## [Author Response]

This study includes several strong points, but also some weaker aspects:– The quality of the work is high and the apparent specificity of the reporter for CDK4/6 is quite surprising. However, the evidence supporting and explaining this specificity is currently not extensive.– While individual cell tracing is certainly valuable, in this case the conclusions about the temporal order largely confirm the conclusions from previous studies that were based on kinase assays, Western blots and IPs from bulk cell populations.– Insight in the kinetics of CDK4/6 activation is important and novel. A limitation here is the sole use of the MCF10a cell line, which lacks the most important CDK4/6 antagonist, p16.– The stress response data is new and interesting. Here, one would have liked to see DNA damage included in the analysis, in particular because it remains unclear to what extent the experimental procedure – time-lapse fluorescence microscopy – induces a "stress" response that contributes to the observed bifurcation in the control cell population.

We thank the reviewers for the overall positive comments. We have completed the requested experiments and, in response to the helpful comments, we also made changes to the overall text and figures for clarification of novel insights and to address different specific issues raised by the reviewers.

As one change in the manuscript, we had a group discussion at the GRC proliferation conference this summer among different users of the cyclin E/A-CDK reporter and agreed upon a term to use in manuscripts. We now explain in the manuscript first the specificity of the reporter (it can collectively measure cyclin E/A-CDK2/1 activity) but then use the term CDK2 reporter in the text as it is typically the main measured activity particularly in G1 and S phase. We thought a similar convention is also useful for the CDK4/6 reporter in this manuscript which can also be partially regulated by CDK2 activity, particularly in S and G2 when CDK2 activity is much higher than in G1 phase.

We agree with the point made above that the manuscript confirms many known aspects of CDK4/6 and CDK2 activity regulation. However, there are also key findings we made about the kinetics of CDK4/6 and CDK2 activity changes that were surprising. We have now reorganized the manuscript to more clearly state some of the expectations and what we learned from the live cell analysis how cells can enter and exit the cell cycle. We also more clearly explain the unexpected increasing persistence in CDK4/6 activity when cells approach S-phase. To improve the overall clarity, we also reorganized the figure panels and, in response to the reviewers points, added the following new data and panels to the manuscript: Figures 3B, 3F and 4F and Figure 1—figure supplements 1A, 2C-E, 3B-C and 4C-D.

Essential revisions:1) The development of a specific CDK4/6 sensor is of great importance, for this study as well as the field of cell cycle and cancer research. The specificity of the reporter should therefore be confirmed beyond any doubt. Currently the validation of the specificity relies entirely on the response to chemical CDK inhibitors. The high specificity of the CDK4/6 inhibitors and the time of addition (after kinase activation) puts some confidence in these experiments. However, the contribution of CDK2 in phosphorylating the CDK4/6 sensor remains a bit uncertain (see points below). Moreover, although a docking site for CDK4/6 has been added, it is unclear why this sensor would not be seen by CDK2-cyclin E/A kinases. Additional experiments are needed to validate and clarify the specificity. First and foremost, it is critical to compare and contrast the combined sensors in CDK4/6 double knockout cells. Further, the protein sequence of the reporter should be provided, and testing additional drugs, in particular against MAP kinases, mutating the Rb binding site, and performing in vitro kinase assays would help characterize and clarify the specificity of the reporter.

The reviewers raise a valid point about reporter specificity and we discussed how to best address this. A potential problem with using CDK4/6 knockout cells is that different groups have shown that cyclin Ds bind to CDK2 when CDK4/6 proteins are missing. In these cells, CDK2 now may potentially functioning similarly to CDK4/6 as a cyclin D regulated kinase. We nevertheless added a number of additional controls characterizing the specificity of the reporter:

–A main point that was also raised the GRC conference on proliferation where we presented this work is the possibility that cyclin E/A-CDK activity may also be needed to phosphorylate the CDK4/6 reporter – in addition to CDK4/6 activity. Such a co-requirement of both kinases has been proposed by some groups for Rb phosphorylation. One way to address this point is to knockout all cyclin Es and As and to show that the reporter can still be regulated in a CDK4/6-dependent manner. Peter Sicinsky kindly provided us with Cre inducible knockout MEFs where we could delete all four cyclin Es and As. As we now show in Figure 1—figure supplement 3B-C, knockout of cyclin Es and As completely abolished the measured CDK2 reporter signal, as expected. However, these cells still have an increased CDK4/6 reporter signal that can be inhibited by CDK4/6 inhibitors, arguing that CDK4/6 activity is sufficient to generate a CDK4/6 reporter signal.

– We now further clarified that the reporter can also be partially phosphorylated by CDK2. In added text and figure panels, we now more clarify show that CDK2 can regulate the reporter, particularly in S and G2 phase when the activity of CDK2 is much higher than in G1 phase. The contribution of CDK2 activity on the CDK4/6 reporter in G1 is a small fraction of the signal and stays below ~20% even towards the end of G1 but is about 35% towards the end of S/G2 (Figure 1E, Figure 1—figure supplement 2B-C). We now explain more clearly that one can almost completely correct for this contribution from CDK2 activity by subtracting a fraction of the CDK2 reporter signal from the CDK4/6 reporter signal. With this correction, inhibition of CDK4/6 at any point in the cell cycle abolishes the mitogen-triggered increase in the CDK4/6 reporter signal (Figure 1—figure supplement 4B-C).

– We would further like to also highlight that we already used in our initial submission 3 structurally distinct CDK4/6 inhibitors that all nearly completely suppress the CDK4/6 reporter signal during G1. Also, both CDK2 selective inhibitors inhibit the CDK2 reporter but have only a small effect on the CDK4/6 reporter in G1. As an additional control, we have also added a panel with titrations of the effect of different types of other kinase inhibitors on the CDK4/6 reporter which supports the notion that the reporter is suitable to measure CDK4/6 activity (Figure 1—figure supplement 2C).

2) The Materials and methods section is much too condensed and needs to be expanded by describing every experiment in detail. The specific parameters that are used in the image analyses and for the statistical analyses need to be stated. The exact nucleotide sequence of the CDK4/6 reporter needs to be described.

We have now expanded the Materials and methods section and added the amino acid sequence of the reporter construct (Figure 1—figure supplement 1A) as well as the requested statistical information.

3) Although the exact timing of reporter activity is highly variable, results clearly show in every single cell that mitogen stimulation first induces CDK4/6 activity, followed on average 2 h later by Cyclin E-CDK activity and after another 3-4h loss of APC/C-CDH1 activity which demarcates S-phase entry. The specificity of the reporters was confirmed by using several CDK inhibitors. Figure 1F shows that CDK2 inhibitors did reduce the activity of the CDK4/6 reporter, but the authors consider this "a negligible effect". What is the explanation? Non-specificity of the CDK2 inhibitors, partial response of the CDK4/6 reporter to CDK2 activity, feedback of CDK2 activity on CDK4/6 activity?

As discussed in the points above and in point 4 below, we have now reorganized the text and added control experiments to better explain how CDK2 activity contributes to the measured reporter signal which is particularly necessary to correct the measured CDK4/6 activity in S and G2 phase.

4) The next paragraph starts with: "The result of adding CDK4/6 inhibitors was different in S-phase where we found an incomplete suppression of the putative CDK4/6 reporter signal." Please make clear which figures need to be compared to arrive at this conclusion (probably the first and third panel of row 2 in Figure 1—figure supplement 2).

We have now clarified in the text how CDK2 activity starts to more significantly contribute to the reporter signal in S and G2 as the activity of CDK2 is ramping up towards mitosis. Since the contribution of CDK2 activity in G1 is relatively small compared to the noise of the reporter measurements, one can in principle use the CDK4/6 reporter without corrections during G1 phase when a majority of users may use it. We now also more clearly explain how a correction using a subtraction of 35% of the CDK2 reporter signal can be used to derive a corrected CDK4/6 activity signal that can be almost completely inhibited by CDK4/6 inhibitor throughout G0/G1/S and G2.

5) Related to this: is the conclusion that the CDK4/6 reporter partially responds to Cyclin E-CDK activity not also supported by the first panel of the second row of Figure 1—figure supplement 2? And if so, wouldn't this also be the case in G1 cells as shown in Figure 2F first panel, second row, requiring some correction in G1 cells as well?

We are now more clear that the correction from the CDK2 activity reporter is applied to the entire cell-cycle time course. We further clarified in the text that this correction is a small fraction of the maximal CDK4/6 reporter signal in G1 phase (<20% is within the noise of the accuracy of measuring the reporter signal), suggesting that a correction can be used but is not absolutely needed in studies that focus on CDK4/6 activity in G0/G1 where CDK4/6 activity is likely functionally most relevant.

6) The concept of "bimodal distribution" of CDK4/6 activity after mitotic exit is questionable. After all, 17% of the cells show a delayed increase in CDK4/6 activity which is not seen in the histograms of Figure 2A as this increase happens just between the chosen time points 2 and 9 h after mitotic exit.The interesting observation is that in the presence of mitogenic stimuli, 69% of cells maintain high CDK4/6 activity whereas cyclin E/A-CDK activity invariably declines after mitotic exit. But which mechanism governs the maintenance or degradation of CDK4/6 activity right after mitosis? Is this controlled by a similar stress response as reported later in the manuscript, and could the frequent exposure to UV light, or other culture conditions, trigger this stress response in part of the control population? It would be very helpful if the authors can clarify this issue.

The reviewers raise a good point and we now address the regulation of CDK4/6 activity after mitosis in different ways. In our working model, that is based in part on our previous study (Yang et al., 2017), it is the competition between cyclin Ds and p21/p27 type CDK inhibitors that decides at anaphase whether or not CDK4/6 activity stays active and Rb is hyperphosphorylated or not. Our previous study showed that, unlike cyclin E and A, cyclin D and p21 are not reset at anaphase, arguing that CDK4/6 activity can stay on at anaphase and help keep Rb stay hyperphosphorylated during and after mitosis. Our observation here that CDK4/6 activity is close to bimodal at mitotic exit (the first time we can reliably measure CDK4/6 activity in daughter cells is about 1-2 hours after anaphase) can explain the previously observed bimodality in Rb hyperphosphorylation and subsequent bifurcation in CDK2 activity at mitotic exit.

The point by the reviewer refers to a second important point about the fraction of cells that first turn CDK4/6 off and then later back on to re-enter the cell cycle. Our study provides new insights into this critical phenomenon that was previously observed by several groups for CDK2 activity (including us, Spencer et al., 2013). In this observation, many cells that exit mitosis first have CDK2 activity off for variable time periods before turning CDK2 activity back on (these cells were termed CDK2^delay^ cells). Our study now shows that there is a close correlation between cells having CDK4/6 on or off and cells either directly activating CDK2 and entering the next cell cycle or keeping CDK2 activity off and exiting to quiescence – or later re-entering the cell cycle. Our study now provides mechanistic insights how this delay is generated. To make this point clearer, we now added a panel of individual activity time courses (Figure 3B) which shows that there is a clearly defined G1 extension after mitosis where CDK4/6 and CDK2 activity are both low for a variable time period. Markedly, this argues that the time cells spend between mitosis and S-phase is controlled by cells entering a transient G0 period of variable duration with low CDK4/6 and CDK2 activity. We previously showed, using only the CDK2 reporter, that the fraction of cells in this transient G0 period increases at lower mitogen stimuli. Our CDK4/6 reporter now adds to this the finding that CDK4/6 activity and CDK2 activity are both jointly low during the transient G0 period and then again sequentially turn on when cells re-enter G1 and the cell cycle by activating CDK2.

7) Apparently, 14 % of cells exiting mitosis do not reactivate CDK4/6 (Figure 2A). What is the fate of these cells? It would be interesting to know whether these cells will die or remain quiescent.

This fraction that exits the cell cycle is the same we and others observed in different previous experiments. We now point this our more clearly in the text. In non-transformed cells, there is always a small fraction of cells in a population that exits to quiescence after mitosis. This fraction increases when mitogen stimuli are reduced and is marked by elevated p21 activity and/or lower cyclin D levels. In published work by us and others and in some unpublished work not related to this study, it has been shown that these cells that exit to quiescence can often re-enter after long delays. Furthermore, the fraction depends not only on the dose of mitogen, but also on the degree of different stresses and nutrient availability, as well as on the degree of contact inhibition. For example, at lower mitogen doses, a larger fraction will exit the cell cycle after mitosis as we described before comparing CDK2^inc^ versus CDK2^low^ cells (Yang et al., 2017). We now added text to make this point about fractional regulation of the cell population at the end of mitosis more clear.

8) Interestingly, the response of cells to several stressors depended on the level of Cyclin E/A-CDK activity: if low, cells responded by lowering CDK4/6 activity and after an initial increase also Cyclin E/A-CDK activity. In part, these different responses appear to reside in different phases of kinase activation: CDK4/6 is at max levels at the time of stress induction, while CDK2-associated kinase activity is rapidly increasing. This should be discussed. Moreover, it is not so clear how in Figure 3A a distinction is made between high and low cyclin E/A-CDK activity. Was that based on the 20 h time point?

Yes, we now clarify this point about how we classify cells in the different panels.

Similarly, for Figure 4—figure supplement 1: how are CDK4/6 high and low defined?

The classification is based on a threshold derived from the bimodal CDK4/6 activity histogram in Figure 3A. We now clarify this point by adding text and making the figure panel more clear.

9) The definitions are important to know in order to appreciate Figure 4—figure supplement 1A and 2A showing that rather than the level of CDK4/6 activity the activity of cyclin E/A-CDK determines whether cells can arrest in response to stress. However, isn't this strange as Figure 1 shows that increasing cyclin E/A-CDK activity is preceded by increasing CDK4/6 activity? After all, the authors conclude that the probability of cells to inactivate CDK4/6 and Cyclin E/A-CDK becomes progressively smaller as the cells progress through G1 phase and approach the G1/S transition. This stage is characterized by high activity of both CDK4/6 and Cyclin E/A-CDK; so why no relation with CDK4/6 activity?

We have now improved these parts of the text and discussion. The main reason for a lack of a correlation in respect to CDK4/6 amplitude is mostly that CDK4/6 activity has typically largely plateaued out in mid to late G1 that, at least as we can measure with the corrected reporter signal. Nevertheless, since CDK4/6 inhibition in G1 is stopping or reversing the increase in CDK2 activity (Figure 1D), it is relevant that CDK4/6 is high in G1 and that stress can reverse the increase in CDK4/6 activity.

10) The panel indication in Figure 1—figure supplement 2 should preferably be adapted to match the legend.

We apologize for the oversight. We have corrected the legend.

11) The use of references is unnecessarily incomplete, some references listed lack important information.

We have now added additional relevant references.

12) The Discussion should be expanded and include a description of how the current observations relate to previously published data.

We have now expanded the Discussion.

[Editors' note: further revisions were suggested prior to acceptance, as described below.]

The experimental part of the work and observations are considered worth publication in eLife. However, the results should be interpreted with more caution, and clearly take into account that the cell line used lacks an important CDK4/6 inhibitor. Moreover, the writing should be broadened, some terminology used more carefully, and the Discussion should critically discuss what can be concluded and what cannot be learned from the current study. Further, more extensive and proper citations of others will be essential. Please see the review below for additional details.

In response to these different points, we have revised the Introduction and Discussion and added in the revised manuscript the requested broader review of the literature on previous work studying the regulation of CDK4/6 and CDK2 activation and Rb phosphorylation and also in more detail the previous characterization of the CDK2 reporter. We also changed comments in the Results and Discussion section to interpret the data with more caution and discuss that the immortalized breast epithelial MCDF10A cell line that we use is lacking the two CDK4/6 inhibitory proteins p16 and p15. To address this point experimentally, we performed experiments where we induced exogenous p16 and showed that this caused a fraction of cells to stay in quiescence but without having an effect on the kinetics of CDK4/6 activation or the time between CDK4/6 and CDK2 activation. We also added data showing similar kinetic changes of CDK4/6 and CDK2 activation after mitosis as in MCF10A cells when we used another commonly used immortalized retinal pigment epithelial model, RPE1 cells. We also added more details on the specificity of small molecule CDK inhibitors used, adjusted the terminology, and added at different places comments that CDK4/6 and CDK2 regulatory mechanisms may differ for different cell types and conditions. The added experiments were performed by two students and a postdoc who are now added as co-authors (with permission from all the authors).

Reviewer #1:[…]Technically, this is a strong study with several important novel findings and observations. Its weakness is a general oversimplification and lack of critical evaluation throughout the text.The explicit request from the reviewers to expand the Discussion and improve the citation of other work has mostly been met with increased self-citations. The manuscript, discussion, and model (Figure 4E) give the impression that the measured activity profiles provide an all-encompassing explanation for how cell cycle entry is regulated. However, as indicated in the previous review, all experiments are performed in MCF-10A cells that lack p16, the most important CDK4/6 antagonist. In fact, MCF-10A cells contain a deletion of the CDKN2A/2B locus (p16, ARF, p15), as well as a MYC amplification (Cowal et al., 2005; Worsham et al., 2006; Kadota et al., 2010). This does not necessarily disqualify the current study – the novel reporter and stress response remain important. However, when considering the CDK4/6 activation profiles and ultrasensitivity of the response, p16 and its absence in the cells studied should be a major factor to consider.

The reviewer makes a good point and we now added data where we compared how CDK4/6 activity changes also in a second commonly used retinal pigment epithelial model, immortalized RPE1 cells. As shown in the added Figure 3—figure supplement 1, the bimodal kinetics of CDK4/6 activation is similar to MCF10A cells and the bimodal increase in CDK4/6 activity can also be inhibited by palbociclib.

We now also clarify in the text that the immortalized MCF10A cells lack p16 and p15. To compare how the kinetics of CDK4/6 activation may change at elevated levels of p16, we now used a DHFR p16 induction system in MCF10A cells. At levels of induction where we see a fraction of cells to be inhibited to enter the cell cycle by p16, we found the those cells that still enter the cell cycle do so with a similar kinetics of the CDK4/6 activity increase and there was a similar delay from the activation of CDK4/6 to the activation of CDK2. This data is now added as Figure 2—figure supplement 1.

In general, critical evaluations and limitations of the experiments should be presented in a transparent way. Contradictions to the previous work, such as the study form Schwartz (Skotheim group), should be fairly cited and not just to draw a parallel with the current work.

i) We understand the reviewer refers to an added characterization by the Skotheim lab of the CDK2 reporter that our group developed. To address this point, we now expanded the section describing the CDK2 reporter in the Introduction:

“We previously introduced a nuclear translocation-based reporter that can monitor the activation of cyclin E-CDK2 in G1 phase (Hahn et al., 2009; Spencer et al., 2013) and different properties of the reporter were characterized in subsequent studies. The reporter can be phosphorylated in vitro by cyclin E-CDK2 or cyclin A-CDK2 activity (Spencer et al., 2013), as well as by cyclin E/A-CDK1 activity (Schwarz et al., 2018), but not by cyclin D-CDK4/6 activity (Spencer et al., 2013). Given that cyclin E prefers CDK2 over CDK1 (Koff et al., 1992), and that cyclin A typically starts to increase at the G1/S transition, this cyclin E/A-CDK2/1 reporter is expected to primarily measure the activity of cyclin ECDK2 during G1 phase. We therefore refer to the reporter here as a “CDK2 reporter”.”

We would like to note that we have also confirmed as part of our *eLife* study here that MEFs lacking cyclin E and A show no CDK2 reporter signal.

ii) The study the reviewer refers to by our Stanford colleagues also reported that primary isolated cells activate CDK2 only after a long delay after mitosis. However, this interpretation may depend on their culture conditions since another study (Moser et al., 2018) showed that apparently identical primary cells as those used by the Skotheim group have the same fast CDK2 increases after mitosis in the hands of this other group. The current view in the field is that different mitogen, contact inhibition and stress signals regulate the fraction of cells that have a fast or a delayed increase in CDK2 activity after mitosis before they enter the next cell cycle. Studies by several groups characterizing how CDK2 activity is regulated after mitosis address this question in more detail and are currently under review. We now comment on the published work on CDK2 activity regulation after mitosis in more detail and added additional citations of relevant work by the Bakal, Skotheim, Spencer and our group how CDK2 activity is regulated after mitosis in the Results and Discussion sections.

Less dramatic, but still absolutely requiring adjustments: the Introduction still contains a fair number of questionable citations (e.g. "Yang et al., 2017" for a general and long established cell cycle regulation mechanism), while other citations are missing (e.g. "the dynamics of CDK4/6……E-CDK2 activation has not been explored"; in fact, the two kinase activities were compared as early as 1994; Meyerson et al. MCB).

Thank you for pointing this out. It was our intention to say that the kinetics between CDK4/6 and CDK2 activation has not been investigated in single cells. Our main point is that there is great heterogeneity when single cells activate CDK2 activity coming out of quiescence which makes it difficult to make a conclusion about the relative kinetics of CDK4/6 and CDK2 activation in single cells from bulk cell data. We now make this clarification in the Introduction and added references for the early studies that used bulk-cell analysis to characterize the time course of cyclin D, E and A expression and cyclin-CDK complex formation.

The inhibitors used should be considered more critically: RO-3306 is called a CDK1 inhibitor and Roscovitin a CDK2 inhibitor, but the differences in affinity are modest and at the concentrations used (60 microM) several other CDKs (including CDK2/A for RO-3306 and CDK1/A,B for Roscovitin) would be expected to be blocked.

This is a good point and we added a more detailed description about their relative specificity and we are now also more precise how we refer to these inhibitors in text and figures.

Also, the inhibitor data in Figure 1—figure supplement 2C would be worth discussing in more detail. Is sequential cyclin E-CDK2, cyclin A-CDK2, cyclin A-CDK1 kinase activity likely responsible for the prolonged increase in CDK2 reporter activation? How does the new insights fit with the recent data that challenge the traditional model for CDK4/6 function (Dyson lab 2019; Dowdi lab 2014).

We now cite the work on the monophosphorylation of Rb by the Dowdy laboratory but we don’t have a good explanation of how their results fits with the observed rapid activation of CDK4/6 after a delay and some of our more recent data. We would like to point the reviewer to a recent paper by our group (Chung et al., 2019) that we now also cite where we show more precisely that MCF10A cells transition from CDK4/6 dependence to independence only at the onset of S-phase when the CDK2 reporter signal has an approximate value of 0.9 (by that time CDK4/6 has been active for about 3 hours and CDK2 activity is sufficiently high to hyperphosphorylate Rb on its own). Upon addition of palbociclib during this time window before CDK2 activity reaches a level of 0.9, Rb becomes still dephosphorylated in less than 15 minutes and CDK2 activity first plateaus out and/or decreases with cells that then go on to enter a quiescent state. At higher levels of CDK2 activity, palbociclib has no more effect and cells keep increasing CDk2 activity and enter S phase. As a speculation that we did not include in the manuscript, one possible interpretation of the Dowdy result is that there is residual CDK4/6 activity in early G1 and cells have Rb monophosphorylated until about 3 hours before S phase when CDK4/6 activation triggers Rb hyperphosphorylation. CDK4/6 activity then remains required for less than 3 hours to keep Rb hyperphosphorylated. Given that this 3 hour window starts about 5-13 hours after mitogen stimulation, and the great variability when cells first activate CDK4/6, it is conceivable that this short window where palboclib has an effect on Rb hyperphosphorylation was missed in the bulk-cell analysis in the Dowdy studies.

There is a discrepancy in the correction factor for the CDK4/6 reporter! The text and cover letter mentions subtracting 35%, while the legend of Figure 1—figure supplement 4 states 25%, and the Materials and methods uses a factor 0.25x.

We thank the reviewer for pointing out this error that we now corrected, the factor used is .35.

Together, the current description of the work is too one-sided, and the interpretation of the results is certainly incomplete and possibly incorrect without considering the absence of p16.

We appreciate the helpful comments by the reviewer. We have now addressed the main concerns by the reviewer by broadening the Introduction and Discussion, by clarifying what specifically can be concluded, and by more clearly stating that CDK4/6 and CDK2 signaling may differ in different cells and for different experimental conditions. We believe that the manuscript has been significantly improved by these recommended changes.